# Association Between Metabolic Syndrome and Risk of Laryngeal Cancer: A Systematic Review

**DOI:** 10.3390/curroncol32110635

**Published:** 2025-11-13

**Authors:** Faizan Bashir, Supriya Peshin, Moniza Rafiq, Sajida Zaiter, Naga Anvesh Kodali, Helia Bazroodi, Moiza Bashir, Lalith Vardhan Choudary, Sakshi Singal

**Affiliations:** 1School of Medicine, Shiraz University of Medical Sciences, Shiraz 71345-3119, Iran; susuzaiter22332@gmail.com (S.Z.); heliabazroodi@yahoo.com (H.B.); moizabashirl@gmail.com (M.B.); 2Department of Internal Medicine, Norton Community Hospital, Norton, VA 24273, USA; supriyapeshin720@gmail.com; 3Department of Physiology, Government Medical College, Baramulla 193101, India; drmoniza2023@gmail.com; 4Department of Internal Medicine, The University of Central Florida, HCA Ocala Hospital, Ocala, FL 35571, USA; anveshkodali@gmail.com; 5Department of Cardiology, The University of Texas MD Anderson Cancer Center, Houston, TX 77030, USA Department of Cardiology, MD Anderson Cancer Centre, Huston, TX 77030, USA; lalithchow@gmail.com; 6Medical Oncology, East Tennessee State University, Johnson City, TN 37604, USA

**Keywords:** metabolic syndrome, MetS, cancer-risk, HNC, laryngeal cancer, endocrine dysfunction

## Abstract

This review looked at whether metabolic syndrome (MetS), a cluster of conditions like high blood pressure, high blood sugar, and abdominal obesity, is linked to laryngeal cancer. After searching major databases, five large studies were included. Three studies from Korea showed that people with MetS had a higher risk of developing laryngeal cancer, even after accounting for smoking and alcohol use. High blood pressure and blood sugar were the most common factors tied to increased risk. However, studies from the UK and the US did not find the same association. Overall, these findings suggest a possible association that may vary by population, highlighting the need for comparable studies outside Korea and for prevention strategies that target metabolic health more broadly.

## 1. Introduction

Laryngeal cancer is one of the most prevalent forms of head and neck cancers (HNCs) [1]. Recent global estimates from GLOBOCAN 2022 report roughly 189,000 new laryngeal cancers and just over 103,000 deaths worldwide in 2022 [2]. Despite the ongoing advancements in the medical and public health sectors, laryngeal cancer remains a significant public health concern worldwide. While traditional risk factors such as smoking, alcohol use, and HPV infection account for most cases, recent evidence suggests that other systemic and potentially modifiable metabolic factors may also contribute to disease risk or progression [3].

Metabolic syndrome (MetS), characterized by central obesity, elevated blood pressure, insulin resistance, and dyslipidemia, is one of the foremost causes of morbidity and mortality worldwide [4]. According to the International Diabetes Federation (IDF), MetS affects approximately one quarter of the global adult population, and this number is growing along with the obesity epidemic [5,6]. Analyses of NHANES from 2011 to 2018 place MetS at roughly four in ten adults in the United States [7].

The underlying pathophysiology of MetS involves interconnected processes of chronic inflammation, oxidative stress, and hormonal dysregulation, collectively fostering a pro-carcinogenic milieu [8]. This has been associated with increased risk of breast, colorectal, and pancreatic cancers [9,10,11,12]. Numerous proposed biological mechanisms substantiate the hypothesis that MetS is associated with cancer. Chronic low-grade inflammation, a hallmark of MetS, promotes tumor growth by secreting pro-inflammatory cytokines such as TNF-α and IL-6, which can stimulate cell proliferation and angiogenesis. Insulin resistance and the consequent hyperinsulinemia can directly stimulate insulin-like growth factor (IGF) pathways, which are significant promoters of cell proliferation and anti-apoptosis. Additionally, adipokines released by adipose tissue, including leptin and adiponectin, have been demonstrated to affect tumor behavior, with heightened leptin levels associated with an increased risk and progression of cancer [9,13,14,15]. Since laryngeal cancer shares certain etiological pathways with other obesity-related malignancies, it is conceivable that these metabolic disturbances also play a role in its pathogenesis.

Several observational studies have investigated the relationship between specific components of MetS and laryngeal cancer risk, producing inconclusive findings. Some population-based studies have indicated a positive correlation between diabetes mellitus and HNC/HNSCC risk [16,17], whereas other studies have identified no such association between MetS and the risk of HNC [18,19]. Likewise, investigations into individual components—such as obesity, elevated fasting glucose, and hypertension—have yielded divergent results, possibly reflecting variations in study design, sample size, or confounder adjustment. To date, the composite effect of MetS as a unified exposure on laryngeal cancer risk has not been systematically synthesized.

The present systematic review aims to comprehensively evaluate existing evidence on the association between metabolic syndrome and the risk of laryngeal cancer, and to determine whether this relationship is consistent across populations and study designs.

## 2. Materials and Methods

This study sought to investigate the association between the risk of laryngeal cancer and metabolic syndrome. Our systematic review was carried out according to preferred reporting items for systematic reviews and meta-analyses (PRISMA) guidelines [20], and was prospectively registered on PROSPERO (registration no. CRD420251107295)

### 2.1. Search Strategy

A comprehensive literature search was conducted in PubMed/Medline, Scopus, and Web of Science based on all entries up to 1 August 2025. The strategy used a combination of MeSH terms and free-text keywords such as “Metabolic Syndrome,” “Insulin Resistance Syndrome,” “Metabolic Abnormalities,” “Hyperglycemia,” “Hypertension,” and “Laryngeal Neoplasms.” These terms were used in each database with the objective of maximizing yield, with an emphasis on specificity. We did not filter for date, language, or type of study to ensure the broadest possible capture of literature. The full search strings for each database are listed in Table 1.

### 2.2. Eligibility Criteria

Studies were eligible if they were observational in design (cohort, case–control, or cross-sectional) and assessed the association between MetS and the risk of laryngeal cancer in adults (≥18 years). Only studies that used an established definition of MetS (e.g., NCEP ATP III, IDF, WHO), reported original data, and had a minimum follow-up or study period of three years were included. Studies reporting on HNCs were also considered eligible, provided that laryngeal cancer was included under HNCs.

We excluded studies focused on treatment, prognosis, survival, or recurrence, as well as reviews, editorials, conference abstracts, animal studies, and studies without clearly defined exposure or outcome definitions.

To handle overlapping cohorts, we included only the most comprehensive or methodologically sound study. If two or more studies using overlapping data addressed substantially different research questions (e.g., baseline vs. time-varying MetS, or subgroup-specific effects), both were retained and interpreted narratively.

### 2.3. Selection of Studies

All search results from the three databases were imported into Zotero referencing manager, and duplicates were manually removed. Two reviewers (S.Z. and M.B.) independently screened all titles and abstracts, followed by full-text review of potentially relevant studies. Disagreements were resolved through discussion, and if necessary, adjudicated by a third reviewer (F.B.). The selection process was summarized in a PRISMA flowchart.

### 2.4. Data Extraction

After selection of studies, data were independently extracted by three reviewers (H.B., F.B. and M.B.) using a predesigned template. Extracted data included study identifiers (authors, year, country), database, design, population age, MetS definition, number and type of MetS components analyzed, effect measures (HR, OR), 95% confidence intervals, statistical significance, and main findings. All discrepancies were resolved through consensus.

### 2.5. Quality Assessment

Methodological quality of included studies was assessed using the Joanna Briggs Institute (JBI) Critical Appraisal Checklists (available from—https://jbi.global/critical-appraisal-tools (accessed on 1 August 2025)), appropriate to each study design. Two reviewers (S.Z. and F.B.) performed the assessments independently.

### 2.6. Data Synthesis

Findings were synthesized narratively. Results were organized thematically across overall MetS diagnosis and its individual components. Direction, association, and significance of associations were systematically summarized. To complement the textual synthesis, we generated schematic forest plots to visually depict the structure and distribution of evidence between MetS components and laryngeal cancer risk. We did not perform a quantitative meta-analysis because the included studies differed in their analytic frameworks. Variations included effect measure (HRs vs. ORs), exposure operationalization (baseline MetS vs. longitudinal change), outcome definition (laryngeal vs. composite head-and-neck cancer), and underlying cohort structure (partially overlapping KNHIS datasets vs. independent Western cohorts).

### 2.7. Software Used

All phases of screening, data extraction, and quality assessment were completed manually without automation tools. Zotero (version 7) referencing software was used for citation management. All figures were generated using Python 3, employing Matplotlib, Pandas, and NumPy for statistical plotting and visualization.

## 3. Results

In August 2025, we conducted a systematic search of PubMed/MEDLINE, Scopus, and Web of Science, initially yielding 788 records. Following duplicate removal, title/abstract screening, eligibility assessment, and full-text screening, five studies met the inclusion criteria (Figure 1). These were large population-based studies carried out using databases from South Korea (KNHIS), the United Kingdom (UK Biobank), and the United States (SEER-Medicare database). Among them, three were KNHIS studies, of the other two studies, one was conducted using UK Biobank and the other was based on the SEER-Medicare database.

Finally, this systematic review included four cohort studies and one case–control study published between 2013 and 2023 that evaluated associations between metabolic syndrome (MetS) and laryngeal or head-and-neck cancers (HNCs); where feasible, laryngeal cancer was treated as a distinct endpoint. Three large Korean reports, all based on the Korea National Health Insurance Service (KNHIS) platform, addressed different analytic questions and sampled populations differently. The 2023 analysis enrolled adults aged ≥20 years who underwent baseline health checks in 2009 (final n = 9,598,085; ~31.6% aged <40) and applied a one-year lag, reporting HNC overall with site-specific (larynx) sub-analyses over approximately nine years of follow-up. Kim et al. (2021) restricted the cohort to individuals > 40 years who completed two sequential examinations (2009 and 2011), producing a selected repeat-exam subcohort (n = 6,757,048; median follow-up 6.4 years) and explicitly modelling changes in MetS status (MetS-developed, MetS-chronic, MetS-recovered, MetS-free). The 2019 study analyzed the national health-check population from 2009–2010, reporting marked age differences between cases and non-cases (mean age 63.3 vs. 47.7 years, respectively) and a distinct case burden and follow-up window.

Because these analyses were conducted within the same KNHIS data infrastructure, some degree of population overlap was possible. In accordance with best-practice guidance for systematic reviews using shared data sources and predefined methods, the KNHIS-derived studies were treated as related but methodologically distinct rather than as independent replications. Key distinguishing features included enrolment age strata and denominators, exposure operationalization (baseline MetS vs. time-varying/change categories), outcome scope (larynx-only vs. composite HNC with subtype analyses), follow-up and lagging choices, and covariate adjustment sets. To avoid inflating the evidentiary weight of a single registry, we did not pool KNHIS-derived estimates quantitatively or summarize them as independent replications. Instead, their results were interpreted qualitatively in relation to one another and to external datasets (UK Biobank, SEER–Medicare) to assess consistency across populations.

Across the KNHIS analyses, the adjusted estimates for MetS and laryngeal (or larynx-specific HNC) were generally positive (≈1.13–1.32), noting potential population overlap, whereas UK Biobank and SEER–Medicare showed null and inverse associations, respectively (Table 2).

Following data extraction and compilation, the results were organized into six domains:(1)Association between MetS and laryngeal cancer(2)Contribution of individual MetS components(3)Effect of component clustering(4)Interaction with smoking and alcohol use(5)Sex-specific and non-linear associations(6)Dynamic changes in MetS status

### 3.1. Association Between MetS and Laryngeal Cancer

As illustrated in Figure 2, the three Korean cohort studies offered the most direct evidence to support a positive association between MetS and laryngeal cancer. Sang-Yeon Kim’s [21] study found Mets was correlated with a 13% increased hazard of laryngeal cancer (HR: 1.13), and the authors observed their reported association remained significant after controlling for confounders including smoking, alcohol intake and physical activity. A dose–response relationship was also observed as the cancer risk increased with increasing numbers of MetS components.

Hyun-Bum Kim [22] and Geun-Jeon Kim’s [23] study had similar findings, with MetS linked to a significantly elevated risk of laryngeal cancer, HR: 1.320 (95% CI: 1.17–1.489) and HR: 1.18; (95% CI: 1.09–1.27), respectively. Moreover, the association persisted for never-smokers and non-drinkers.

In contrast, Stott-Miller [19], using a US-based case–control dataset, found an overall inverse association between MetS and head and neck squamous cell carcinoma (HNSCC) (OR: 0.81; 95% CI: 0.78–0.85). though anatomical subsites were not disaggregated.

Jiang [18], using UK Biobank data, found no significant association between MetS and incident head and neck cancer (HR: 1.05; 95% CI: 0.90–1.22). However, numerous individual components of MetS were independently associated with increased risk, as described below.

**Table 2 curroncol-32-00635-t002:** Baseline characteristics of the included studies assessing the association between MetS and laryngeal/HNC. Studies were conducted using databases of South Korea, UK and The USA between 2013–2023. The table summarizes study design, population details, number of laryngeal cancer cases, age distribution, definition of metabolic syndrome applied, main findings, and methodological quality (JBI assessment).

Study, Year	Country/Database	Study Design	Study Period	Population	Laryngeal Cases	Age Range	MetS Definition	Main Findings	JBI Critical Appraisal
Sang-Yeon Kim et al., (2019) [21]	South Korea/KNHIS	Retrospective cohort	2009–2015	23 M	5322	MetS: >60 years No MetS: >40 years	IDF criteria (≥3 of: waist ≥ 85 cm females/males ≥ 90 cm, BP ≥ 130/85, FBG ≥ 100, TG ≥ 150, HDL < 40 mg/dL for men and <50 mg/dL for women)	• Mean age was higher in individuals with laryngeal cancer. • MetS associated with a 13% ↑ hazard of laryngeal cancer (HR: 1.13; 95% CI: 1.03–1.25). • Increasing number of MetS components correlated with higher risk. • MetS identified as an independent risk factor for laryngeal cancer.	10/12
Geun-Jeon Kim et al., (2023) [23]	South Korea/KNHIS	Retrospective cohort	2009–2018	9.6 M	2972	>20 years	IDF criteria (≥3 of: waist ≥ 85 cm, BP ≥ 130/85, FBG ≥100, TG ≥ 150, HDL <40)	• Participants with MetS had a 1.06-fold ↑ risk of HNC • MetS was significantly associated with increased risk of laryngeal cancer • Elevated FBG & BP were significantly associated with increased HR for HNC • HNC risk remained elevated in never- or former-smokers and non- or mild drinkers.	10/12
Hyun-Bum Kim et al., (2021) [22]	South Korea/KNHIS	Retrospective cohort	2009–2018	6.75 M	1350	>40 years	IDF criteria (≥3 of: waist ≥ 85 cm, BP ≥ 130/85 or the use of antihypertensive medication, FBG ≥ 100 or the use of hypoglycemic agent, TG ≥ 150, HDL < 40 mg/dL for men or <50 mg/dL for women)	• Individuals with MetS had a 1.13-fold higher hazard of developing laryngeal cancer. • Laryngeal cancer risk increased with the number of MetS components. • Abdominal obesity conferred the highest risk of laryngeal cancer among components.	12/12
Jiang et al., (2021) [18]	China/UK Biobank	Prospective cohort study	2006–2015	474 K	103	>50 years	AHA/NHLBI criteria (≥3 of: BMI < 30 or waist ≥ 94/80 cm [M/F], BP ≥ 130/85, FBG ≥ 100 or specific treatment, TG ≥ 150 or specific treatment, HDL < 40/50[M/F])	• MetS was not associated with ↑ HNC risk. • ↑ number of MetS components showed no risk. • Hyperglycemia independently correlated with ↑ HNC risk. • U-shaped associations for HDL-C and waist circumference with HNC risk. • CRP ≥ 1.00 mg/dL → ↑ HNC risk.	10/12
Stott-Miller et al., (2013) [19]	USA/SEER-Medicare	case–control study	1994–2007	14 K cases 42 k controls	5666	68–99 years	NCEP-ATP III criteria (≥3 of: elevated waist circumference/central obesity, dyslipidemia, hypertension, and impaired fasting glucose or diabetes)	• Moderate inverse association between MetS and HNSCC risk (OR = 0.81). • Inverse association strongest for overweight & dyslipidemia. • Impaired FBG and hypertension showed weak inverse associations (OR = 0.90; OR = 0.95, respectively.	10/10

↑ = increase; → = leads to.

### 3.2. Contribution of Individual MetS Components

Consistent across studies, however, were fasting glucose and hypertension as important risk factors for laryngeal cancer. Geun-Jeon Kim found HRs of 1.13 (95% CI: 1.05–1.21) for fasting glucose and 1.21 (95% CI: 1.12–1.31) for blood pressure. Similar findings were reported in earlier Korean studies, where elevated blood glucose and blood pressure were associated with increased risk [21,22,23]. In Jiang’s UK Biobank cohort [18], elevated fasting glucose was associated with increased HNC risk following full adjustment (HR: 1.22, 95% CI: 1.02, 1.45), no statistically significant risk was observed with high blood pressure (HR: 1.00, 95% CI: 0.82, 1.20). Interestingly, one study [19] found a weak inverse association (closer to null) between high blood pressure and HNSCC risk (OR =  0.95; 95% CI 0.90–0.99). Likewise, impaired fasting blood glucose also showed negative association (OR = 0.90; 95% CI: 0.86–0.94).

Lipid abnormalities demonstrated more mixed results. Geun-Jeon Kim, controlling for confounders, found elevated triglycerides to be associated with increased risk of HNC. Additionally, these results are consistent with earlier work on this database. Both the 2019 and 2021 studies consistently identified elevated triglycerides and low HDL-C as factors contributing to an increased incidence of laryngeal cancer [21,22,23]. Jiang’s study identified a U-shaped relationship between HDL-C and HNC cancer in men, suggesting potential nonlinearity or effect modification by sex. This study also included dyslipidemia for TG as a risk factor in the unadjusted model [18]. Alternatively, dyslipidemia was found to be strongly inversely associated with HNSCC risk in Stott-Miller et al. [19].

Central obesity showed no correlation with HNC risk (Jiang et al., 2021; HR: 1.04; 95% CI: 0.90, 1.21). However, waist circumference demonstrated considerable risk (HR: 1.09; 95% CI: 1.01–1.18) largely among males [18]. Only one Korean study [22] found a statistically significant HR for abdominal obesity in relation to laryngeal cancer (HR: 1.242; 95% CI: 1.067–1.445). The other two Korean studies reported no significant association. Stott-Miller [19] reported a strong negative association for overweight (OR: 0.69; 95% CI: 0.64–0.74).

The associations and directions of effect for individual MetS components are presented in Figure 3a–d.

### 3.3. Cumulative Effect of MetS Component Clustering

Sang-Yeon Kim looked at the different three- and four-component combinations of MetS to see which clusters were associated with the highest hazard ratios for increased risk [21]. Notably:Abdominal obesity, high blood pressure, and impaired fasting glucose combined were associated with a 32.5% higher risk (HR 1.32; 95% CI: 1.13–1.55).High triglycerides, impaired fasting glucose, and low HDL together conferred a 35.7% higher risk (HR 1.36; 95% CI: 1.11–1.66).A four-component cluster including high blood pressure, elevated triglycerides, impaired fasting glucose, and low HDL was linked to a 33.7% higher risk (HR 1.34; 95% CI: 1.17–1.53).Abdominal obesity, high blood pressure, elevated triglycerides, and low HDL combined showed a 22.6% higher risk (HR 1.23; 95% CI: 1.03–1.47).

One study analyzed the impact of elevated C-reactive protein (CRP), specifically levels greater than 1.00 mg/dL, independently increased HNC risk by 21% when compared to lower CRP levels (<1.00 mg/dL) (HR: 1.21; 95% CI: 1.02–1.43). This same study also investigated the joint effect of CRP and MetS on HNC risk [18], reporting:Elevated CRP in MetS individuals was linked to a 29% greater HNC risk (HR: 1.29; 95% CI: 1.05–1.58).Even without MetS, elevated CRP still resulted in 22% higher HNC risk (HR: 1.22; 95% CI: 1.02–1.47).Joint effect between CRP and MetS was undetectable, where p-interaction > 0.05.

### 3.4. Influence of Smoking and Alcohol Use

Two Korean studies examined whether the association between MetS and laryngeal cancer was independent of smoking and alcohol use. In the 2019 study, the increased risk linked to MetS persisted even among never-smokers and non-drinkers, indicating that metabolic dysfunction may play a carcinogenic role irrespective of these traditional exposures [21].

In the 2021 study, elevated laryngeal cancer risk was observed across the MetS chronic, developed, and recovery groups—even among individuals who were never or former smokers, or who reported no or mild alcohol intake. Ex-smoker status was associated with increased risk only in the MetS chronic group. Notably, current smoking and heavy alcohol consumption did not further elevate risk in any of the MetS groups [22].

In contrast, Stott-Miller [19] found more pronounced inverse associations among tobacco use (OR: 0.72; 95% CI: 0.67–0.77)

### 3.5. Dynamic Changes in Metabolic Syndrome Status

One study specifically looked at the relationship between changes in MetS status and the risk of laryngeal cancer. The MetS chronic group had the highest risk of laryngeal cancer (HR 1.42; 95% CI: 1.229–1.641). The second highest risk was seen in the MetS developed group (HR 1.296; 95% CI: 1.093–1.537), followed by the MetS recovery group (HR 1.220; 95% CI: 1.008–1.476). In the MetS chronic group, risk was associated with being middle-aged, male, a never or ex-smoker, a non- or mild alcohol drinker, and having no dyslipidemia. This group also showed a link to laryngeal cancer that was independent of diabetes or hypertension status. In the MetS developed group, being middle-aged, male, a never-smoker, a non-drinker, having no dyslipidemia, and having a low BMI were tied to an increased risk of laryngeal cancer. Interestingly, in the MetS recovery group, being male, a never smoker, and having no dyslipidemia were also associated with a higher risk [22].

### 3.6. Sex-Specific and Nonlinear Associations

Another complex level of analysis relates to the sex-specific associations between individual MetS components and laryngeal cancer. Jiang [18] conducted restricted cubic spline analyses and noted significant U-shaped associations for waist circumference in women and HDL-C in men. In this case, risk may be conferred by metabolic markers at both extremes, rather than in a simple linear fashion.

For example, while increased waist circumference generally conferred higher risk among women (HR: 1.47; 95% CI: 1.15–1.89 for ≥93.16 cm), both too low (<1.26 mmol/L) and too high (≥1.26 mmol/L) HDL-C levels in men were linked to elevated HNC risk (HR: 1.19; 95% CI: 1.06–1.34 for high HDL-C). This study also examined U-shaped associations with other parameters, such as blood glucose and diastolic blood pressure that were too high and too low; however, no correlation was found.

Being male was found to be a risk factor for developing laryngeal cancer in the MetS chronic, MetS developed, and MetS recovery groups [22]. In the 2019 KHNIS study [21], males were reported to have higher incidence of laryngeal cancer compared to women 93.16% vs. 6.84%.

### 3.7. Quality Assessment

The JBI assessment indicated that all included studies were of high methodological quality, with scores ranging from 10/12 to 12/12 for cohort studies and 10/10 for the case–control study. Most studies provided clear definitions of exposures and outcomes and adjusted for major and relevant confounders such as smoking and alcohol use, though unmeasured factors (e.g., diet, HPV status, or adequate BMI status) and reliance on administrative data may still pose a risk of residual bias.

## 4. Discussion

This systematic review investigated the correlation between MetS and laryngeal cancer risk by analyzing large population-based research from three nations. Our review identified a positive association between MetS and laryngeal cancer in Korean cohorts after adjustment for smoking and alcohol, while Western cohorts did not replicate this pattern. The most reliable risk predictors were elevated fasting glucose and blood pressure, and having more components of MetS was associated with greater risk of laryngeal cancer. Overall, the evidence supports a role for long-term metabolic health in laryngeal cancer risk.

The three Korean cohort studies provided strong evidence of an independent and statistically significant positive association between MetS and laryngeal cancer risk. This association was especially notable among never-smokers and non-drinkers, indicating that metabolic dysfunction constitutes an independent carcinogenic pathway, separate from conventional lifestyle-related exposures. The observed dose–response relationship, in which cancer risk escalated in direct correlation with the number of MetS components, reinforces the evidence for a causal association.

These findings may align with emerging evidence from other HNC sites. In a 2022 study, Choi et al. found that those with MetS had an 11.3% higher risk of oral cavity cancer compared to healthy controls, with risk increasing linearly with each additional MetS component [24]. Likewise, in a Catalonian cohort, López-Jiménez et al. (2024) identified MetS components linked to earlier cancer diagnosis. Their Cox regression analyses revealed that low HDL-C (HR 1.46, 95% CI 1.41–1.52) and hyperglycemia (HR 1.40, 95% CI 1.37–1.44) were the strongest predictors of overall cancer risk, providing insight into which metabolic disturbances may be most oncogenic [25].

The association between MetS and cancer risk is biologically plausible and consistent with known metabolic and inflammatory pathways that link insulin resistance, dyslipidemia, and chronic low-grade inflammation to carcinogenesis [26].

Among the individual MetS components, hyperglycemia and hypertension were the most consistent risk factors across studies. Chronic hyperglycemia promotes carcinogenesis through formation of advanced glycation end-products (AGEs), which subsequently bind to receptors on macrophages and endothelial cells, initiating reactive oxygen species (ROS) generation and eventual DNA damage [15]. This mechanistic pathway is well-supported by many epidemiological studies associating diabetes and different cancers [9]. This pathway is further depicted in Figure 4, which provides an overview of the principal biological mechanisms through which chronic hyperglycemia drives carcinogenesis.

The carcinogenic effects of hyperglycemia, however, are not restricted to laryngeal cancers. Remschmidt et al. demonstrated significantly higher rates of glucose metabolism disorders in oral cancer patients compared to controls (*p* < 0.00001) [27]. An umbrella review of systematic reviews and meta-analyses has confirmed direct links between diabetes mellitus and increased oral, oropharyngeal, head and neck, and nasopharyngeal cancer risk [28]. Becker et al. also found that long-term metformin consumption was associated with decreased risk of laryngeal cancer, suggesting pharmacologic glycemic control may provide protective features [29].

Hypertension likewise showed consistent associations with cancer risk. Seo et al. identified hypertension as a “remarkable risk factor” for the development of laryngeal cancer (HR 1.23), indicating that people with untreated hypertension are at the greatest risk for malignancies such as oral, laryngeal, and esophageal, whereas Connaughton and Dabagh identified hypertension as a cancer initiator in their meta-analysis. Recent UK Biobank data, for example, further supported hypertension as an independent cancer risk factor, though the exact mechanisms connecting elevated blood pressure to carcinogenesis require further exploration through future large-scale investigations [30,31,32].

The contrasting results from Western populations raise important methodological and population-related issues. The Korean studies consistently showed positive associations, while the UK Biobank research found no association between MetS and HNC, and the SEER-based study indicated an inverse association. These discrepancies may reflect differences in baseline metabolic profiles, healthcare systems, genetic susceptibility, and analytical frameworks, including model adjustment strategies.

Methodological differences, particularly the adjustment for BMI, may obscure real metabolic effects. Because BMI is both a MetS component and an independent cancer risk factor, adjusting for it may overcontrol for a mediator and attenuate the observed association. Population-related issues like patterns of body composition, dietary habits, and genetic polymorphisms that affect metabolic capacity, may impact the strength and nature of the association.

## 5. Limitations

Our review has several limitations. Although most studies adjusted for smoking and alcohol use, residual confounding from other lifestyle or occupational exposures, as well as HPV status, cannot be excluded. Outcome classification also varied, with some studies grouping laryngeal cancer within broader head and neck cancer categories, raising the possibility of misclassification. A further consideration is that much of the available evidence comes from Korean cohorts, with potential overlap across datasets, which may limit the degree of independent replication and reduce generalizability to other populations with different genetic and environmental contexts. Despite these limitations, our review synthesizes comprehensive evidence on the potential link between metabolic syndrome and laryngeal cancer, highlighting consistent biological pathways.

## 6. Conclusions and Future Directions

This systematic review identified a reproducible positive association between metabolic syndrome (MetS) and laryngeal cancer in Korean cohorts derived from the KNHIS platform, although the direction and magnitude of effect differed across populations. Analyses from the UK Biobank and SEER–Medicare databases yielded null and inverse associations, respectively, underscoring that the observed relationship may be population-specific rather than universally causal. Taken together, the current evidence base supports a context-dependent link between long-term metabolic dysfunction and laryngeal carcinogenesis, warranting cautious interpretation and further validation.

Several uncertainties remain unresolved. The biological relevance of specific MetS components, their clustering patterns, and their cumulative or synergistic effects on oncogenic pathways are not yet well characterized. Moreover, most existing studies employ binary case definitions of MetS, which may obscure dose–response gradients or temporal transitions in metabolic status. Future research should adopt trajectory-based or severity-weighted frameworks, integrate biomarkers of insulin resistance, inflammation, and adipokine signaling, and assess whether metabolic interventions can modify cancer risk in a prospective manner.

Cross-population differences also highlight the need for harmonized, multi-ethnic longitudinal studies using consistent exposure definitions, follow-up windows, and analytic strategies. Such work will be essential to distinguish context-specific epidemiologic patterns from shared mechanistic pathways linking systemic metabolic dysregulation to laryngeal cancer risk.

## Figures and Tables

**Figure 1 curroncol-32-00635-f001:**
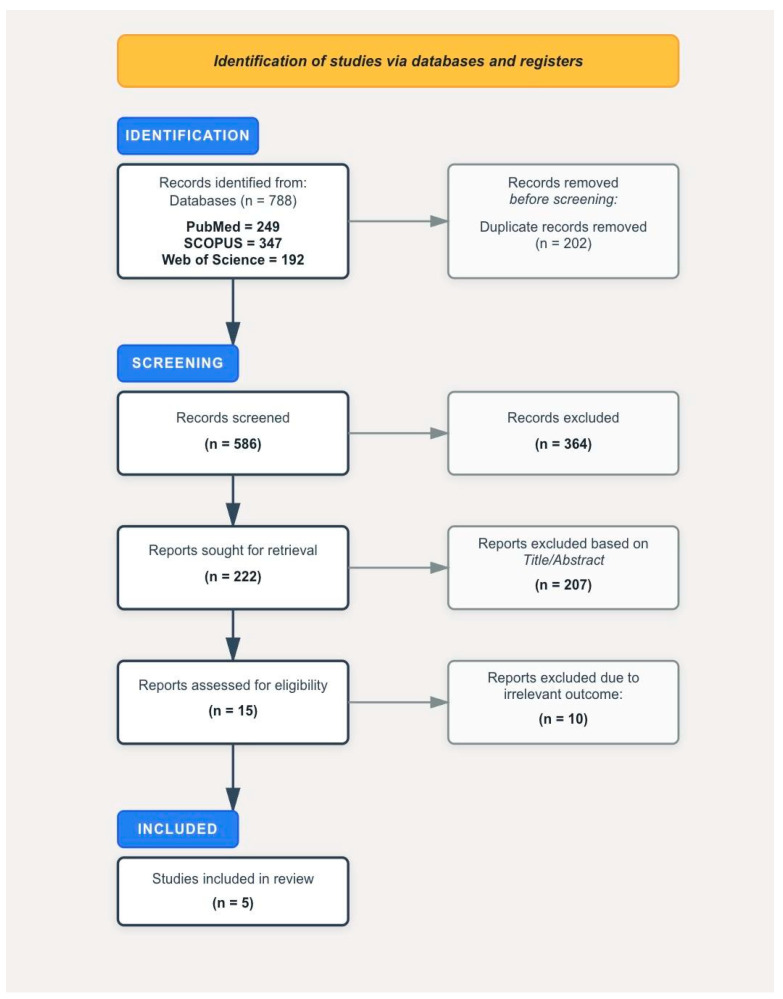
PRISMA flowchart documenting the selection process of studies.

**Figure 2 curroncol-32-00635-f002:**
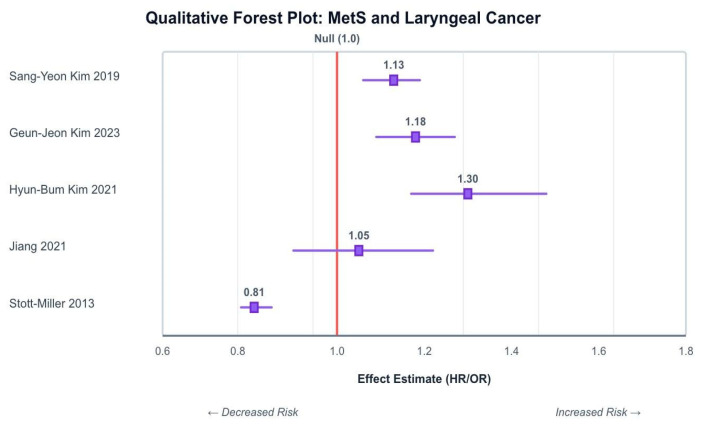
Schematic forest plot summarizing reported associations between MetS and laryngeal/HNC across included studies. The Korean cohort consistently indicates a positive association [18,19,21,22,23].

**Figure 3 curroncol-32-00635-f003:**
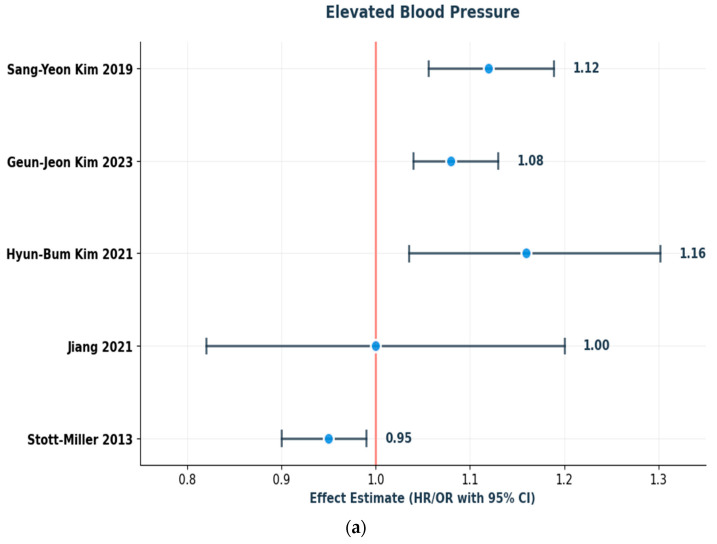
(**a**–**d**): Schematic forest plots showing the direction of effect for individual MetS components in relation to laryngeal/HNC risk: (**a**) elevated blood pressure, (**b**) elevated fasting glucose, (**c**) increased waist circumference, and (**d**) HDL-C. Reported effect estimates (HRs/ORs with 95% CIs) are presented as reported in the original studies [18,19,21,22,23].

**Figure 4 curroncol-32-00635-f004:**
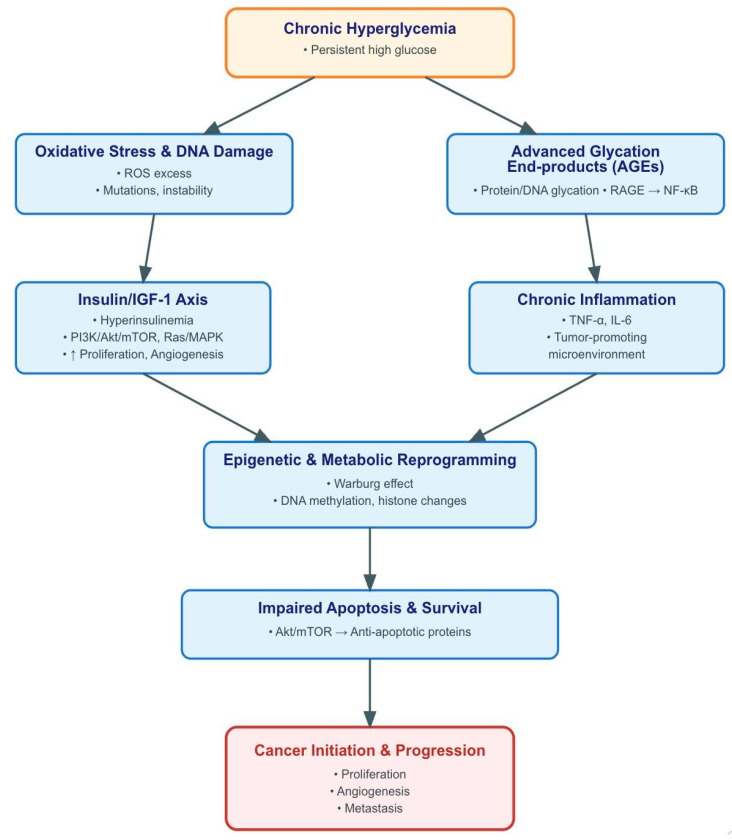
Brief overview of pathogenetic mechanisms linking chronic hyperglycemia to cancer development. Sustained hyperglycemia promotes tumorigenesis through multiple, interrelated pathways. Excess glucose enhances insulin and insulin-like growth factor-1 (IGF-1) signaling, activating phosphoinositide 3-kinase (PI3K)/protein kinase B (Akt)/mammalian target of rapamycin (mTOR) and Ras/mitogen-activated protein kinase (MAPK) cascades that drive proliferation and angiogenesis. Concurrently, hyperglycemia induces mitochondrial overproduction of reactive oxygen species (ROS), leading to DNA damage and genomic instability, while the formation of advanced glycation end-products (AGEs) activates the receptor for advanced glycation end-products (RAGE) and downstream nuclear factor kappa-light-chain-enhancer of activated B cells (NF-κB) signaling, amplifying inflammation. The chronic inflammatory milieu, characterized by cytokines such as tumor necrosis factor-alpha (TNF-α) and interleukin-6 (IL-6), sustains a tumor-promoting microenvironment. In parallel, high glucose availability fosters metabolic reprogramming (Warburg effect) and epigenetic alterations, including DNA methylation and histone modifications, further supporting malignant transformation. These processes collectively impair apoptosis and enhance survival signaling, culminating in cancer initiation and progression.

**Table 1 curroncol-32-00635-t001:** Database specific search strategies used in PubMed/MEDLINE, Scopus, and Web of Science during the literature review. Search strings combined “Metabolic Syndrome” terms AND “Laryngeal Cancer” or “Head and Neck Neoplasms” terms using Boolean operators. Synonyms within each concept (e.g., “Metabolic Syndrome X,” “Insulin Resistance,” “Syndrome X”) were joined with OR, and the two concept blocks were joined with AND. The asterisk * was used as a truncation symbol in to capture word variants in the search terms.No restrictions were set on language or publication year; eligibility refinements (study design, exposure definition, outcome type) were conducted manually during screening.

Database	Search Strategy	Records Found	Searched Until
PubMed/MEDLINE	(“Metabolic Syndrome”[Mesh] OR “Metabolic Syndrome X” OR “Insulin Resistance Syndrome” OR “Syndrome X” OR “MetS” OR “metabolic abnormalities” OR “metabolic disorder *” OR “hyperglycemia” OR “hypertension” OR “dyslipidemia” OR “central obesity”) AND (“Laryngeal Neoplasms”[Mesh] OR “laryngeal cancer” OR “head and neck cancer” OR “larynx cancer” OR “cancer of the larynx” OR “laryngeal carcinoma” OR “laryngeal squamous cell carcinoma” OR “LSCC” OR “laryngeal tumor *” OR “laryngeal malignan *”)	249	1 August 2025
Scopus	TITLE-ABS-KEY (“metabolic syndrome” OR “metabolic syndrome x” OR “insulin resistance syndrome” OR “syndrome x” OR “mets” OR “metabolic abnormalities” OR “metabolic disorder *” OR “hyperglycemia” OR “hypertension” OR “dyslipidemia” OR “central obesity”) AND TITLE-ABS-KEY (“laryngeal cancer” OR “larynx cancer” OR “cancer of the larynx” OR “laryngeal carcinoma” OR “laryngeal squamous cell carcinoma” OR “LSCC” OR “laryngeal tumor *” OR “laryngeal malignan *”)	347	1 August 2025
Web of Science	TS = (“Metabolic Syndrome” OR “Metabolic Syndrome X” OR “Insulin Resistance Syndrome” OR “Syndrome X” OR “MetS” OR “metabolic abnormalities” OR “metabolic disorder *” OR hyperglycemia OR hypertension OR dyslipidemia OR “central obesity”) AND TS = (“Laryngeal Neoplasms” OR “head and neck cancer” OR “laryngeal cancer” OR “larynx cancer” OR “cancer of the larynx” OR “laryngeal carcinoma” OR “laryngeal squamous cell carcinoma” OR LSCC OR “laryngeal tumor *” OR “laryngeal malignan *”)	192	1 August 2025

## Data Availability

All data generated or analyzed during this study are derived from previously published studies, which are cited within the article. No new datasets were created or analyzed for this review.

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
