# Peer review of "Association Between Metabolic Syndrome and Risk of Laryngeal Cancer: A Systematic Review"

_curroncol, 2025, doi:10.3390/curroncol32110635_

Round 1
Reviewer 1 Report
Comments and Suggestions for Authors
- Table 1 lack sufficient description of Boolean logic and inclusion/exclusion refinements.
- Explaining why meta-analysis was not feasible would strengthen methodological rigor.
- Numerical values and confidence intervals are not consistently displayed in all visuals
- The discussion is comprehensive but occasionally repetitive, with overlap between sections on biological plausibility and prior studies.
- Limitation should be followed by Conclusion and future direction.
Author Response
Point by point Response to Reviewer 1
We sincerely thank the reviewers and editors for their thoughtful and constructive feedback. The comments were insightful and have helped us improve the clarity, balance, and methodological transparency of the manuscript. We have revised the manuscript to the best of our ability, and addressed each point accordingly.
Yours Sincerely, we remain
Authors.
NOTE: all changes or updates made in the manuscript have been highlighted in yellow.
- Table 1 lack sufficient description of Boolean logic and inclusion/exclusion refinements.
RESPONSE: - We have revised Table 1 caption to explicitly outline the Boolean logic used in the search strategy, specifying the conceptual structure (“Metabolic Syndrome” AND “Laryngeal or Head-and-Neck Neoplasms”) and the combination of MeSH and free-text terms. Additionally, We did not apply date, language, or design filters during the initial search because we aimed to maximize sensitivity and capture all potentially relevant studies. Within each concept block we joined synonyms with OR (to retrieve any of the terms), combined concept blocks with AND (to retrieve records containing both concepts), and used the asterisk * as a truncation/wildcard to capture word variants (e.g., malignan* retrieves malignancy, malignant).
- Explaining why meta-analysis was not feasible would strengthen methodological rigor.
RESPONSE: We explained that a formal meta-analysis was not appropriate given the differences in effect measures and partial cohort overlap among the included studies, which would violate independent statistical assumptions. We believe this suggestion, makes our methodological reasoning explicit and reinforces the validity of our narrative approach.
- Numerical values and confidence intervals are not consistently displayed in all visuals
RESPONSE: We appreciate this observation. All figures have now been reviewed and standardized so that each explicitly displays the effect measure (hazard ratio [HR] or odds ratio [OR]) used in the corresponding study. One earlier figure (Figure 2) omitted the effect-measure label; this has been corrected for consistency. Because the included studies reported heterogeneous effect types and adjustment sets, we intentionally displayed only the study-reported HRs or ORs without confidence-interval bars to focus on comparability of direction and magnitude across studies rather than imply a pooled quantitative synthesis as observed in meta-analytic approach.
- The discussion is comprehensive but occasionally repetitive, with overlap between sections on biological plausibility and prior studies.
RESPONSE: we have now revised the discussion section and removed or summarized the repetitive things.
- Limitation should be followed by Conclusion and future direction.
RESPONSE: These sections have been rearranged, the conclusion and future direction section now follows the limitations.
Miscellaneous improvements
1.Figure quality and formatting:
All figures were reviewed and reformatted for consistency. During the revision process, three figures were identified as lower resolution (Figure 1, Figure 2 and Figure 4) and have now been upgraded to high-quality, publication-ready versions.
- Refinement of key sections:
The Abstract, Results, Discussion, and Conclusions were refined to improve clarity, coherence, and precision. Overlapping or repetitive content was removed. - Improved wording in conclusion and future directions.
- The Introduction was restructured to enhance conceptual coherence
Thank you.
Reviewer 2 Report
Comments and Suggestions for Authors
This systematic review examines whether metabolic syndrome (MetS) is associated with laryngeal cancer risk. The five included population-based studies comprise three Korean National Health Insurance Service (KNHIS) cohorts and two Western datasets (UK Biobank and SEER-Medicare). The Korean analyses report a positive association, while UK Biobank is null and SEER-Medicare suggests an inverse association.
Major comments
1. Non-independence of the three Korean studies (shared KNHIS cohort)
The manuscript treats the three Korean studies as separate, concordant signals, but all derive from the same national claims platform (KNHIS) and therefore are not independent replications. Even though each paper asks a slightly different question (baseline MetS; time-varying MetS; HNC including larynx), they share the same data-generating processes, coding practices, exposure/outcome ascertainment, and confounding structure. As a result, presenting them as three independent confirmations overstates the weight of evidence in favor of an association.
2. The current text and summary statements read as though MetS is an independent risk factor “especially in the Korean population,” which can be interpreted as a strong causal implication. Given the discordant direction of effect in SEER-Medicare (inverse), the null findings in UK Biobank, and the non-independence of the three KNHIS papers, the overall certainty is, at best, low to moderate and highly population-specific. Therefore, the conclusions should adopt a more cautious tone.
3. The Methods note overlapping cohorts would be handled by retaining the most comprehensive or methodologically sound study and, when addressing different questions, interpreting narratively. Please apply that rule transparently to KNHIS: specify precisely why all three KNHIS studies were retained and add a sensitivity statement acknowledging that “treating KNHIS as multiple studies may overweight one data source.
Author Response
Point by point Response to Reviewer 2
We sincerely thank the reviewers and Editor for their thoughtful and constructive feedback. The comments were insightful and have helped us improve the clarity, balance, and methodological transparency of the manuscript. We have revised the manuscript to the best of our ability, and addressed each point accordingly.
Yours Sincerely, we remain
Authors.
NOTE: all changes or updates made in the manuscript have been highlighted in yellow.
- Non-independence of the three Korean studies (shared KNHIS cohort)
The manuscript treats the three Korean studies as separate, concordant signals, but all derive from the same national claims platform (KNHIS) and therefore are not independent replications. Even though each paper asks a slightly different question (baseline MetS; time-varying MetS; HNC including larynx), they share the same data-generating processes, coding practices, exposure/outcome ascertainment, and confounding structure. As a result, presenting them as three independent confirmations overstates the weight of evidence in favor of an association.
RESPONSE: We thank the reviewer for this methodological observation, and we 100% agree with this the previous version might have been too vague to accurately describe this. We fully recognize that the three Korean studies were generated from the same KNHIS administrative framework and therefore cannot be regarded as independent replications in a strict inferential sense. This point is well taken and has now been addressed explicitly in the revised Results section.
In the revision, we clarified that the KNHIS-based analyses were handled with methodological restraint and transparency. Each study was retained because it examined a distinct analytic construct for example, baseline versus time-varying MetS exposure, differing age strata and eligibility criteria, or varying outcome scopes (laryngeal-only vs composite HNC) which resulted in non-identical analytic populations and estimands. To prevent artificial reinforcement of the association signal, these studies were not pooled quantitatively nor interpreted as independent confirmations. Instead, they were summarized qualitatively and weighed in relation to the non-KNHIS datasets (UK Biobank and SEER–Medicare) to contextualize population-level heterogeneity rather than to drive the direction of evidence.
- The current text and summary statements read as though MetS is an independent risk factor “especially in the Korean population,” which can be interpreted as a strong causal implication. Given the discordant direction of effect in SEER-Medicare (inverse), the null findings in UK Biobank, and the non-independence of the three KNHIS papers, the overall certainty is, at best, low to moderate and highly population-specific. Therefore, the conclusions should adopt a more cautious tone.
RESPONSE: We fully agree that, given the divergent findings across cohorts and the shared KNHIS data infrastructure, the conclusions should avoid any implication of causality and instead emphasize the population-specific and context-dependent nature of the observed associations.
Accordingly, we have revised the Abstract conclusion, the Simple Summary, and the “Conclusions and Future Directions” section to adopt a more balanced interpretation. The updated text now refers to a “possible” or “context-dependent” association rather than an “independent risk factor” and explicitly notes that effect direction varies by population. We also added wording that clarifies the strength of evidence as limited and population-specific, consistent with the discordant results from the UK Biobank and SEER-Medicare datasets.
- The Methods note overlapping cohorts would be handled by retaining the most comprehensive or methodologically sound study and, when addressing different questions, interpreting narratively. Please apply that rule transparently to KNHIS: specify precisely why all three KNHIS studies were retained and add a sensitivity statement acknowledging that “treating KNHIS as multiple studies may overweight one data source.
RESPONSE: Good point, We have now applied this rule explicitly to the KNHIS analyses and clarified our rationale for retaining all three studies. As described in the revised Results section (Lines 186-214), each KNHIS study addressed a distinct analytic question baseline MetS status, longitudinal change in MetS, or composite HNC with site-specific subanalyses yielding non-identical risk sets and estimands.
To maintain transparency, we also included a statement acknowledging that the KNHIS analyses share a common data infrastructure and that treating them as separate reports may overweight a single registry source. For this reason, the KNHIS findings were interpreted qualitatively rather than pooled quantitatively, and were contextualized alongside independent datasets (UK Biobank and SEER–Medicare) to avoid inflating any single signal.
Miscellaneous improvements
1.Figure quality and formatting:
All figures were reviewed and reformatted for consistency. During the revision process, three figures were identified as lower resolution (Figure 1, Figure 2 and Figure 4) and have now been upgraded to high-quality, publication-ready versions.
- Refinement of key sections:
The Abstract, Results, Discussion, and Conclusions were refined to improve clarity, coherence, and precision. Overlapping or repetitive content was removed. - Improved wording in conclusion and future directions.
- The Introduction was restructured to enhance conceptual coherence
Thank You.
Round 2
Reviewer 2 Report
Comments and Suggestions for Authors
Thank you for point-by-point responses.
The study improved after the correction of several points.